# Identification and Functional Analysis of a Novel *CTNNB1* Mutation in Pediatric Medulloblastoma

**DOI:** 10.3390/cancers14020421

**Published:** 2022-01-14

**Authors:** Lide Alaña, Caroline E. Nunes-Xavier, Laura Zaldumbide, Idoia Martin-Guerrero, Lorena Mosteiro, Piedad Alba-Pavón, Olatz Villate, Susana García-Obregón, Hermenegildo González-García, Raquel Herraiz, Itziar Astigarraga, Rafael Pulido, Miguel García-Ariza

**Affiliations:** 1Pediatric Oncology Group, Biocruces Bizkaia Health Research Institute, Plaza de Cruces 12, 48903 Barakaldo, Spain; idoia.marting@ehu.eus (I.M.-G.); piedad.albapavon@osakidetza.eus (P.A.-P.); olatz.villatebejarano@osakidetza.eus (O.V.); susana.garciaobregon@osakidetza.eus (S.G.-O.); mariaiciar.astigarragaaguirre@osakidetza.eus (I.A.); miguelalejandro.garciaariza@osakidetza.eus (M.G.-A.); 2Biomarkers in Cancer Unit, Biocruces Bizkaia Health Research Institute, Plaza de Cruces 12, 48903 Barakaldo, Spain; carolinenunesxavier@gmail.com (C.E.N.-X.); rafael.pulidomurillo@osakidetza.eus (R.P.); 3Department of Tumor Biology, Institute for Cancer Research, Oslo University Hospital, The Norwegian Radium Hospital, 0310 Oslo, Norway; 4Department of Pathology, Hospital Universitario de Cruces, Osakidetza, Plaza de Cruces 12, 48903 Barakaldo, Spain; laura.zaldumbideduenas@osakidetza.eus (L.Z.); lorena.mosteirogonzalez@osakidetza.eus (L.M.); 5Department of Genetics, Physical Anthropology and Animal Pathology, Faculty of Science and Technology, University of the Basque Country, UPV/EHU, 48940 Leioa, Spain; 6Department of Physiology, Faculty of Medicine and Nursing, Campus de Leioa, University of the Basque Country, UPV/EHU, 48940 Leioa, Spain; 7Oncohematology Pediatric Unit, Department of Pediatrics, Hospital Universitario de Valladolid, C/Ramon y Cajal n°3, 47003 Valladolid, Spain; hgonzalez@saludcastillayleon.es (H.G.-G.); rherraiz@saludcastillayleon.es (R.H.); 8Pediatric Oncohematology Unit, Pediatrics Department, Hospital Universitario Cruces, Osakidetza, Plaza de Cruces 12, 48903 Barakaldo, Spain; 9Pediatrics Department, Faculty of Medicine and Nursing, University of the Basque Country, UPV/EHU, Plaza de Cruces 12, 48903 Barakaldo, Spain; 10IKERBASQUE, Basque Foundation for Science, 48009 Bilbao, Spain

**Keywords:** medulloblastoma, *CTNNB1*, β-catenin, mutation

## Abstract

**Simple Summary:**

We have analyzed a panel of 88 pediatric medulloblastoma tumors for exon 3 mutations from the *CTNNB1* gene and identified eight missense point-mutations and one in-frame deletion. We describe and functionally characterize a novel *CTNNB1* in-frame deletion (c.109-111del, pSer37del, ΔS37) found in a pediatric patient with a classic medulloblastoma, WNT-activated grade IV (WHO 2016). To the best of our knowledge, this mutation has not been previously reported in medulloblastoma, and it is uncertain its role in the disease development and progression. Our analysis discloses gain-of-function properties for the new ΔS37 β-catenin variant.

**Abstract:**

Medulloblastoma is the primary malignant tumor of the Central Nervous System (CNS) most common in pediatrics. We present here, the histological, molecular, and functional analysis of a cohort of 88 pediatric medulloblastoma tumor samples. The WNT-activated subgroup comprised 10% of our cohort, and all WNT-activated patients had exon 3 *CTNNB1* mutations and were immunostained for nuclear β-catenin. One novel heterozygous *CTNNB1* mutation was found, which resulted in the deletion of β-catenin Ser37 residue (ΔS37). The ΔS37 β-catenin variant ectopically expressed in U2OS human osteosarcoma cells displayed higher protein expression levels than wild-type β-catenin, and functional analysis disclosed gain-of-function properties in terms of elevated TCF/LEF transcriptional activity in cells. Our results suggest that the stabilization and nuclear accumulation of ΔS37 β-catenin contributed to early medulloblastoma tumorigenesis.

## 1. Introduction

Medulloblastoma (MB) is the primary malignant tumor of the Central Nervous System (CNS) most common in pediatric populations and represents 15–20% of the total primary CNS tumors in this age group [1]. MB arises in the posterior fossa, usually from the cerebellar vermis in the roof of the fourth ventricle. MBs have a marked propensity to metastasize via cerebrospinal fluid (CSF) pathways, and evidence of such metastatic spread is present in up to 35% of cases at diagnosis [1,2]. Over the past ten years, the development of genome-wide techniques has yielded significant insights into the biology and genomic complexity of MB that have allowed the definition of specific entities and has provided the basis for the identification of target genes involved in the initiation and progression of the tumors. Based on this information, the World Health Organization (WHO) classification of 2016, in addition to the histopathological classification, stratified patients in the following MB groups: WNT-activated, SHH-activated (*TP53* mutated or *TP53* wild-type), and non-WNT/SHH (Group 3 and Group 4) [2]. Each group presents distinctive patterns of DNA methylation, gene expression profiles, genomic alterations, and clinical prognosis [3,4,5,6,7,8,9]. Robust and validated methods are available to allow us both a precise and differential diagnosis of these MB entities according to the updated WHO classification. A set of different immunohistochemical markers, including β-catenin, YAP1, p75-NGFR, OTX2, and p53, combined with targeted sequencing and copy number assessment such as FISH analysis for *MYC* genes led us a precise assignment of patients for risk-adapted stratification [10].

Recent studies have identified new driver mutations assigned to most patients belonging to Group 3 and 4 MB molecular subgroups, whose genetics and biology remain less clear [11]. New molecular subtypes were differentially enriched for specific drivers as KBTBD4 and PRDM6 in these subgroups [12]. Moreover, through large-scale methylation and transcriptome profiling, new subgroups have emerged resulting in the latest WHO 2021 CNS tumor classification; four subgroups of SHH-MB and eight subgroups of non-WNT/SHH MBs [13].

Immunohistochemically, WNT-activated MB shows the nuclear accumulation of β-catenin protein as a surrogate biomarker for WNT activation caused by *CTNNB1* activating mutations or mutations in *APC* or other genes encoding components of the WNT signaling pathway [14,15,16,17]. The *CTNNB1* gene, that maps to the short arm of chromosome three (3p22.1) and consists of 15 exons encoding 781 amino acids, encodes the β-catenin protein which is a subunit of the cadherin/catenin multiprotein complex. Previous studies have found *CTNNB1* mutations in association with several types of cancer, with a hot spot region targeting the exon three (encoding residues 5–80 from β-catenin). Mutations at exon three usually decrease the phosphorylation-dependent ubiquitination of β-catenin. This renders more stable β-catenin proteins that accumulate in the nucleus and display pro-tumorigenic properties [18,19,20,21]. Remarkably, in pediatric MB, β-catenin exon three mutations have been found to correlate with favorable patient outcomes (greater than 90% overall survival) across independent clinical trials-based biological studies [14,22]. WNT-activated MB are associated with the loss of an entire copy of chromosome six in the majority of cases, while this sub-group is independent of chromosome 17 aberrations, the most common chromosomal alterations detected in MB [14,23]. Although WNT-activated MB tend to display classic or large-cell/anaplastic histology and arise in older children, these tumors cannot be readily distinguished from other tumors that show equivalent clinical and histological features and require identification at the molecular level [14,22,24].

The precise identification of WNT-activated MB tumors is important because of their prognosis in the pediatric age, which provides guidance for the inclusion of patients into ongoing therapeutic trials aiming to prove that reduction of treatment intensity is possible in these patients [14,22] (e.g., the European SIOP PNET5 MB trial (ClinicalTrials.gov Identifier: NCT02066220). In the setting of such clinical trials, it is widely recommended to use two independent methods for reliable identification of these patients, such as immunohistochemistry for β-catenin and sequencing of *CTNNB1* exon three or alternative molecular methods (Nanostring RNA profiling, methylation classifiers) [2,25].

Thus, comprehensive evaluation of pediatric MB is necessary to inform about risk stratification and treatment. In this study, we report the histological analysis and *CTNNB1* mutational status of 88 pediatric MB. We report one novel *CTNNB1* MB mutation, c.109-111del (p.Ser37del, ΔS37). Functional analysis of ΔS37 β-catenin protein variant disclosed gain-of-function properties in cells.

## 2. Materials and Methods

### 2.1. Tumor Material

Tumor samples were collected from 88 pediatric MB patients between the years 2016 and 2021 from different Spanish Hospitals. Ethics approval was obtained from the Ethics committee of Euskadi (CEIm-E), and the study was conducted in accordance with the Declaration of Helsinki. Haematoxylin and eosin-stained slides were reviewed for confirmation, diagnosis, and assignment of histological type. All patients were aged <18 years at the time of diagnosis.

### 2.2. Histology and Immunohistochemistry

Tumor tissue samples from all the patients were fixed in buffered 4% formalin, embedded in paraffin, and stained according to standard procedures. The immunohistochemistry followed standardized methods in the Hospital Universitario de Cruces Pathology Lab, Spain. We used β-catenin (14, Roche Tissue Diagnostics, Roche Ltd.), p53 (Bp53-11, Roche Tissue Diagnostics, Roche Ltd., Oro Valley, AZ, USA), YAP1 (EP1674Y, Abcam Ltd.), GAB1 (H7, Santa Cruz Biotechnology, Inc., Dallas, TX, USA), and OTX2 (1H12C4B5, Thermo Fisher Scientific, Waltham, MA, USA) as primary antibodies. Immunohistochemical staining was performed in an automated immunostainer (BenchMark ULTRA Slide Stainer IHC/ISH System, Roche Tissue Diagnostics, Ventana Medical Systems, Roche Ltd. Oro Valley, AZ, USA).

### 2.3. Fluorescence In Situ Hybridization

The Fluorescence In Situ Hybridization (FISH) molecular test was performed in all samples using commercially available probes from ZytoVision for *MYC* amplification (ZytoLIght SPEC MYC/CEN8 Dual Colour Probe) and *NMYC* amplification (ZytoLIght SPEC MYCN/2q11 Dual Colour Probe), and the Meta Systems probe was used for monosomy 6 (Triple Colour XL 6q21/6q23/6cen). FISH studies were performed on formalin-fixed, paraffin-embedded (FFPE), 3 μm thick sections. Fluorescence staining was visualized with an Olympus BX61-microscope (Olympus, Volketswil, Switzerland) equipped with DAPI, SpectrumGreen, and SpectrumOrange filters. At least 50 nonoverlapping nuclei were analyzed. For the amplification, multiple copies of the green signal or green signal clusters should be observed. For monosomy 6, loss of one green and/or one orange signal should be observed.

### 2.4. Tumor DNA Isolation and Mutational Analyses of the CTNNB1 Gene

Genomic DNA from frozen tumors was extracted with All Prep^®^ DNA/RNA Mini kit (Qiagen, Hilden, Germany), and DNA from FFPE tissue sections were extracted with QIAamp^®^ DNA FFPE Tissue kit. Mutations in exon 3 of the *CTNNB1* gene were detected in genomic DNA obtained from frozen tumors and FFPE tissues using Sanger direct method. The PCR reactions were carried out with the following primer pair: *CTNNB1*_F: 5′-GATTTGATGGAGTTGGACATGG and *CTNNB1*_R: 5′-TGTTCTTGAGTGAAGGACTGAG [26]. Samples were amplified using KAPA Taq polymerase, through 35 cycles on a Mastercycler personnel (Eppendorf, Hamburg, Germany) at 95 °C denaturation for 30 s, 60 °C annealing for 30 s, and 72 °C extension for 30 s. PCR reactions were performed in a volume of 20 µL with 100 ng of genomic DNA in a PCR 10X buffer with MgCl_2_ (KAPA BIOSYSTEMS), 10 mM of dNTPs mix, 10 pmol of each primer, and 0.5 unit of KAPA Taq polymerase (KAPA BIOSYSTEMS). After PCR amplification, products were loaded onto 2% agarose gel. The single and double strands of the PCR products were visualized by Midori Green Advance DNA stain (Nippon Genetics Europe GmbH, Düren, Germany). The resulting PCR products were purified using spin columns (QIAquick PCR purification kit; Qiagen, Hilden, Germany), and sequenced on an ABI 373A sequencer (Applied Biosystems, Waltham, MA, USA).

Variants were visually inspected and identified using the biological sequence alignment editor BioEdit 7.2 and the Basic Local Alignment Search Tool (BLAST). Once the variant was identified, it was annotated using the ENSENBL Genome Browser 105 (transcript ENST00000349496.1). Variant nomenclature was based on the genome version GRCh38.p13 (hg38).

### 2.5. Plasmids and Cell Culture

U2OS human osteosarcoma cells were grown in Dulbecco’s Modified Eagle Medium (DMEM, Sigma) supplemented with 10% fetal bovine serum (FBS, Sigma, St. Louis, MO, USA) and 2 mM Ala–Gln (#G8541, Sigma). Mammalian expression plasmid, pRK5 β-catenin, has been previously reported [27]. pRK5 β-catenin variants were made by PCR oligonucleotide site-directed mutagenesis, as described [28]. Plasmids used for luciferase assays were pEZX-PG04 SEAP (GeneCopoeia, Rockville, MD, USA), pGL4.27 luc2P (#E8451; Promega, Madison, WI, USA), and pGL4.49 luc2P TCF/LEF (#E4611; Promega). U2OS cells were transfected using Effectene (Qiagen) transfection reagent following manufacturer’s recommendation.

### 2.6. Protein Eextracts, Immunoblot Analysis, and Antibodies

Cell were harvested 72 h post-transfection for immunoblot analysis. Cells were lysed in ice cold M-PER lysis buffer (Thermo Fisher, Waltham, MA, USA) supplemented with phosphatase inhibitor PhosSTOP (Roche) and complete Mini protein inhibitor cocktail (Roche), followed by centrifugation at 15,200× *g* for 10 min and collection of the supernatant. Whole cell protein extracts were resolved on 4–10% SDS-PAGE (NuPAGE, Invitrogen) under reducing conditions and transferred to PVDF membranes (Immobilon-FL, Millipore, Burlington, MA, USA). Immunoblot analysis was performed as previously reported [29]. The antibodies used were anti-β-catenin (#19807, Cell Signaling, Danvers, MA, USA), and anti-α-tubulin (CP06, Millipore). The secondary antibody was rabbit anti-mouse (DAKO, Denmark). Band quantification was made with ImageJ.

### 2.7. Luciferase Reporter Assay

1 × 10^5^ cells were plated in 12-well plates and transfected on the following day as explained above. Intracellular luciferase activity was measured 48 h post-transfection according to Luciferase Reporter Assay kit (Promega). Luminescence was measured using TD-20/20 Luminometer (Turner Design, San Jose, CA, USA). Media was harvested for measuring secreted alkaline phosphatase (SEAP) using Secrete-Pair Dual Luminescence Assay kit (#SPDA-D100; GeneCopoeia) according to the manufacturer’s recommendations. The extracellular SEAP values were used to normalize for differences in transfection efficiency. Absorbance was measured on Victor3 microplate reader (PerkinElmer, Waltham, MA, USA).

### 2.8. Statistical Analysis

The two-tailed student t test was used to evaluate statistical significance (GraphPad). *p* values of *p* < 0.05 were considered significant and marked in the results with an asterisk (*). Error bars represent data as mean ± standard deviation from at least 3 independent experiments.

## 3. Results

### 3.1. Histological and Molecular Subgroups of the Pediatric MB Cohort

Our cohort consisted of 88 pediatric MB aged <18 at diagnosis. We classified this cohort into different molecular subgroups based on the histological and immunohistochemical features of the tumors. Examples of distinct patterns of the histological defined entities and the immunohistochemistry of different markers in each subgroup are represented in Appendix A, respectively.

The majority of the tumors, 69.3% (61/88), exhibited classic histology, whereas desmoplastic/nodular and anaplastic/LCA accounted for 13.6% (12/88) and 17.1% (15/88), respectively. The WNT-activated subgroup comprised 10.2% (9/88) of the cases. Non-WNT/SHH MBs was enriched in our cohort, making up 70.5% (62/88) of the cases, whereas SHH-MBs with TP53 wt (wild-type) accounted for 17% (15/88), and SHH-MBs with TP53 mut (mutated) accounted for 2.3% (2/88) (Figure 1A and Appendix A).

The four molecular subgroups varied in histological distribution. Histological type was associated with molecular group: the majority of the non-WNT/SHH MB had classic histology and 10% of all the cases had large cell/anaplastic histology. All the WNT-activated MB subgroup presented classic histology except for one case, which had large cell/anaplastic histology. For the SHH-MB TP53 wt subgroup, all the cases showed desmoplastic/nodular histology except two cases that were classic MB. Finally, all the cases of SHH-MB TP53 mut exhibited large cell/anaplastic histology (Figure 1B and Appendix A).

### 3.2. CTNNB1 Mutation Analysis

We performed Sanger sequencing of *CTNNB1* exon three in the tumor DNA from all patients. All WNT-activated MB tumors had *CTNNB1* exon three mutations, accounting for seven different mutations (Table 1), and displayed nuclear β-catenin. The samples from the other subgroups of MB did not show any mutation in *CTNNB1* exon three, and displayed non-nuclear β-catenin. Five cases showed mutations in the same residue but different transitions: two of our patients presented the same mutation c.98C>T (p.Ser33Phe). The other two cases showed mutations in c.98C>A (p.Ser33Tyr) and one case in c.98C>G (p.Ser33Cys). Three mutations resulted in amino acid substitutions in codons 32 and 34, c.94G>T (p.Asp32Tyr), c.95A>G (p.Asp32Gly) and c.100G>A (p.Gly34Arg) (Table 1).

One of the mutations found in *CTNBB1* resulted in the in-frame deletion of a TCT codon and the loss of Ser37 residue in the variant β-catenin protein (c.109-111del (p.Ser37del)) (Figure 2). To the best of our knowledge, this mutation has not been described in MB (Table 1). Figure 3A illustrates the N-terminal β-catenin residues targeted by mutation in our cohort of pediatric MB, including the novel variant described in this study. In Figure 3B, the frequency of amino acid substitutions in this region of β-catenin in pediatric MB is shown, as retrieved from https://pecan.stjude.cloud (accessed on 1 December 2021). Since the function of β-catenin p.Ser37del (ΔS37) is uncertain, we studied this variant more in detail.

### 3.3. Clinical, Pathological and Molecular Characteristics of the Tumor with the Novel β-Catenin Variant c.109-111del (p.Ser37del)

The novel mutation found in codon 37 was from a 9-year-old, previously healthy, male child presented to the hospital emergency room with holocraneal cephalea every day with two weeks of evolution. Evaluation showed bilateral papilledema, without nausea, or visual or auditive alteration. A craniospinal Magnetic Resonance Imaging (MRI) was performed, having a cerebellar tumor centered on the IV ventricle with expansion thereof. The first diagnosis conclusion was a posterior fossa tumor. The affectation of the cranial exit foramen, in this case left Luschka, is more typical of ependymoma, however, its basal hyperdensity on CT suggests that a MB is the first possibility. After surgery there was a post-surgical residual tumor <1.5 cm^2^. CSF on day +15 showed the presence of atypical cells, suggesting metastatic spread. Frozen and formalin-fixed, paraffin-embedded (FFPE) tumor samples from the patient were included in the analysis. Basic hematoxylin-eosin and reticulin stain, immunohistochemistry and FISH analyses were performed. After MB diagnosis, treatment was administered (similar to per COG ACNS-0332) with initial radiotherapy with concomitant vincristine and carboplatin, and then maintenance chemotherapy with cisplatin, vincristine, and cyclophosphamide, beginning in 21 January 2019 ending on 2 October 2019. In 20 December 2019, another craniospinal MRI was performed, and no contrast enhancement suggestive of relapse was observed. There are no signs suggestive of tumor spread through the CSF.

Histological sections displayed tumoral fragments with large tissue artefact. Valuable areas were densely cellular with a diffuse growth pattern (Figure 4A) and no reticulin net enhancement (Figure 4B). Tumor cells were immature and discohesive, with ill borders, unrecognizable cytoplasm, and a large ovoid nucleus with membrane moulding. There was high mitotic and apoptotic activity and no anaplastic or large cell changes were noticeable. (Figure 4C,D). The immunoprofile showed patchy nuclear stain for β-catenin (Figure 5A), diffuse nuclear stain for YAP1 (Figure 5B) and OTX2 (Figure 5C), and negativity for GAB1 (Figure 5D) and p53 (Figure 5E). FISH analysis displayed monosomy six without *MYC* and *MYCN* amplifications. These results agree with a grade IV WNT-activated MB (WHO 2016).

### 3.4. Functional Characterization of the Novel β-Catenin Variant c.109-111del (p.Ser37del)

In order to functionally characterize the novel p.Ser37del (ΔS37) β-catenin variant, we engineered a c.109-111del β-catenin human cDNA, which was used for transient transfections in U2OS human osteosarcoma cells. Expression of the ΔS37 β-catenin variant was monitored by an immunoblot using anti-β-catenin antibody, in comparison with wild-type β-catenin and the variant S33C, frequently found in pediatric MB (Figure 6A). As shown, both ΔS37 and S33C β-catenin variants displayed a significantly higher protein expression than wild-type β-catenin, suggesting an increase in their protein stability. This is consistent with previous reports showing that mutations in *CTNNB1* encoding β-catenin S33 residue inhibits the proteosomal degradation of the protein [24]. Next, we performed TCF/LEF transcriptional activity assays in U2OS cells overexpressing wild-type and β-catenin variants, as a measure of β-catenin biological activity in cells. ΔS37 and S33C β-catenin variants significantly increased the TCF/LEF transcriptional activity at higher levels than wild-type β-catenin (Figure 6B). In summary, our analysis discloses gain-of-function properties for the novel ΔS37 β-catenin variant in cells.

## 4. Discussion

The adscription of MB tumors to molecular subgroups has become of critical importance for the inclusion/exclusion criteria in the new MB clinical trials. In this study, based on immunohistochemistry, FISH, and the mutational status of *CTNNB1* gene, we have classified a cohort of 88 pediatric MB patients according to the consensus MB molecular subgroups WNT-activated, SHH and non-WNT/SHH. The molecular and histological classification of our MB cases fit with the distribution found in previous studies [4,10], with non-WNT/SHH MB being the more frequent molecular subgroup, followed by the SHH and WNT-activated MB subgroups. Classic histology was prevalent in both non- WNT/SHH and WNT-activated MB subgroups (Figure 1).

The WNT-activated MB subgroup is associated with a good prognosis (>95% survival at five years in pediatric patients), is rarely metastatic at diagnosis (5%–10% of cases), and rarely recurs [4,23]. By contrast, non-WNT MB (SHH, Group 3 and Group 4) are characterized by metastatic disease, increased rates of recurrence, and intermediate/poor overall survival [30]. All MB WNT-activated tumors from our study carried exon three *CTNNB1* mutations, including one novel MB *CTNNB1* mutation c.109-111del (p.Ser37del). Tumor-associated mutations in *CTNNB1* exon three target the N-terminal phosphorylation domain of β-catenin protein, resulting in impaired phosphorylation by GSK-3β, increased β-catenin protein stabilization, and nuclear β-catenin protein accumulation [17,19,20,21,31].

The *CTNNB1* gene encodes β-catenin, which is a central component of the cadherin/catenin complex and participates in cell adhesion and gene transcription regulation mediated by the WNT signaling pathway. Previous studies have found the association of *CTNNB1* mutations with several cancers [18,32,33,34,35,36]. Most of these mutations target *CTNNB1* exon three, and mainly result in substitutions of residues Ser33, Ser37, Thr41, and Ser45 which are phosphorylation targets of GSK3β and CK1α kinases [37,38,39]. These mutations prevent β-catenin phosphorylation, resulting in β-catenin stabilization and nuclear translocation and accumulation, followed by altered transcription of WNT/β-catenin pathway target genes [40,41,42,43]. β-catenin nuclear accumulation indicates WNT pathway activation in MB and affects 18–30% of cases overall [14,44]. A strong correlation is observed between *CTNNB1* mutations and β-catenin nuclear accumulation in MB clinical samples, with *CTNNB1* mutations detected in 60–70% of β-catenin immunohistochemistry nucleopositive cases [14,16,23,44,45]. In our study, we detect mutations in *CTNNB1* and nuclear β-catenin accumulation in all the WNT-activated MB cases analyzed.

The *CTNNB1* mutation described in this work (c.109-111del (p.Ser37del)), creates the deletion of a TCT codon, resulting in the loss of Ser37, a residue phosphorylated by GSK3-3β. To the best of our knowledge, the *CTNNB1* mutation c.109-111del has not been previously found in MB, and it has only been reported in one case of adult hepatocellular carcinoma (COSMIC and cBioPortal databases).

Phosphorylation of β-catenin N-terminal domain has been postulated to occur in a sequential manner (Ser45-Thr41-Ser37-Ser33), although evidence suggests that phosphorylation at Ser33, Ser37, or Thr41 can occur in the absence of phosphorylation at Ser45 [46]. Phosphorylation of these Ser/Thr residues target β-catenin for ubiquitination and proteasomal degradation upon interaction with the ubiquitin ligase β-TrCP [47,48].

In this study we found three mutations resulted in amino acid substitutions in codons 32 and 34, c.94G>T (p.Asp32Tyr), c.95A>G (p.Asp32Gly), and c.100G>A (p.Gly34Arg), suggesting an influence on the accessibility of the Ser 33 for the GSK-3β kinase, preventing its phosphorylation.

Ser37 mutations comprised about 10% of the known ß-catenin mutations, with Ser37Phe, Ser37Cys, and Ser37Ala being the more frequent substitutions. It has been recently described as small molecules that bind to non-phosphorylated or Ser37Ala mutant β-catenin and enhance their binding to β-TrCP, representing a therapeutic strategy to target proteins defective in ligase binding, ubiquitylation, and degradation [49,50]. Our results suggest that mutation p.Ser37del results in altered β-catenin phosphorylation, increased protein stabilization, and nuclear accumulation, contributing to tumorigenesis by the transcriptional activation of gene targets of the WNT/β-catenin pathway.

By contrast, non-WNT MB (SHH, and non-WNT/SHH: Group 3 and Group 4) are characterized by metastatic disease, increased rates of recurrence, and intermediate/poor overall survival [30]. WNT-activated MB rarely metastasize and represent the only MB subgroup in which metastasis is not indicative of a poor prognosis. It has been suggested that WNT signaling may contribute to their remarkable response to standard therapy [51]. Despite being metastatic as diagnosis, the favorable clinical evolution of the WNT-activated MB patients, despite being metastatic at diagnosis, supports that relevance of β-catenin status as an independent marker of favorable clinical outcome [14]. This has raised the question of whether these tumors can be cured with minimal or no radiation and low toxicity chemotherapy [52]. It has been postulated that WNT-activated MB has improved survival because of an impaired blood-brain barrier, suggesting that chemotherapy is an important mainstay of therapy. Paracrine signals driven by mutant β-catenin in WNT-activated MB induce an aberrant fenestrated vasculature that permits the accumulation of high levels of intra-tumoral chemotherapy and a robust therapeutic response [53]. This suggests that a much larger range of drugs could be used to treat WNT-activated MB.

The WNT signaling pathway regulates the development and maturation of the CNS, but its activation may also promote the neoplastic transformation of this tissue [54]. A recent study treating MB xenografts with a WNT agonist, provides a rational therapeutic option in which the protective effects of WNT-activated MB may be augmented in Group three and four MB. This supports further investigation on the context-dependent tumor suppressive role of WNT signaling in MB [51].

## 5. Conclusions

We report nine mutations in exon three of the *CTNNB1* gene from WNT-activated medulloblastoma tumors from a cohort of 88 pediatric medulloblastoma patients. A novel exon three *CTNNB1* mutation targeting codon 37 is described, which has not been previously reported in MB (c.109-111del (p.Ser37del, ΔS37)). We disclosed gain-of-function properties in cells for the novel ΔS37 β-catenin protein variant.

## Figures and Tables

**Figure 1 cancers-14-00421-f001:**
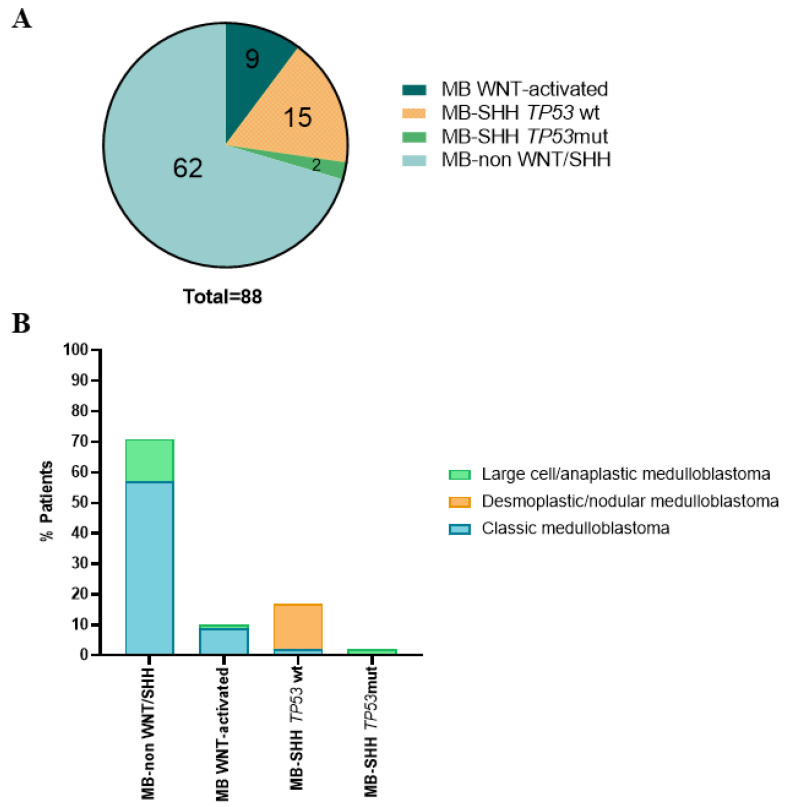
Molecular groups of pediatric medulloblastomas. (**A**): Non-WNT/SHH group are more than 60 cases of the cohort (62/88), Group SHH *TP53*wt are 15 cases (15/88), Group WNT-activated 9 cases (9/88) and Group SHH *TP53*mut, 2 cases (2/88). (**B**): The four groups showed differences in histological distributions.

**Figure 2 cancers-14-00421-f002:**
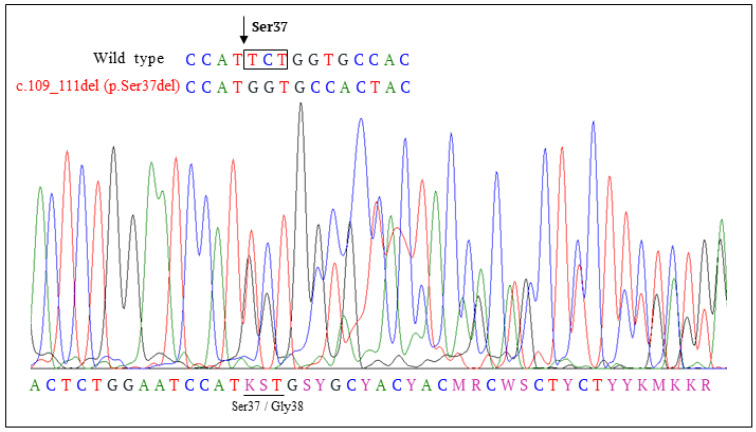
DNA sequencing of *CTNNB**1* exon 3 from a tumor specimen. Sanger DNA sequencing showed the deletion c.109-111del, resulting in deletion of a TCT codon in the tumor sample (p.Ser37del).

**Figure 3 cancers-14-00421-f003:**
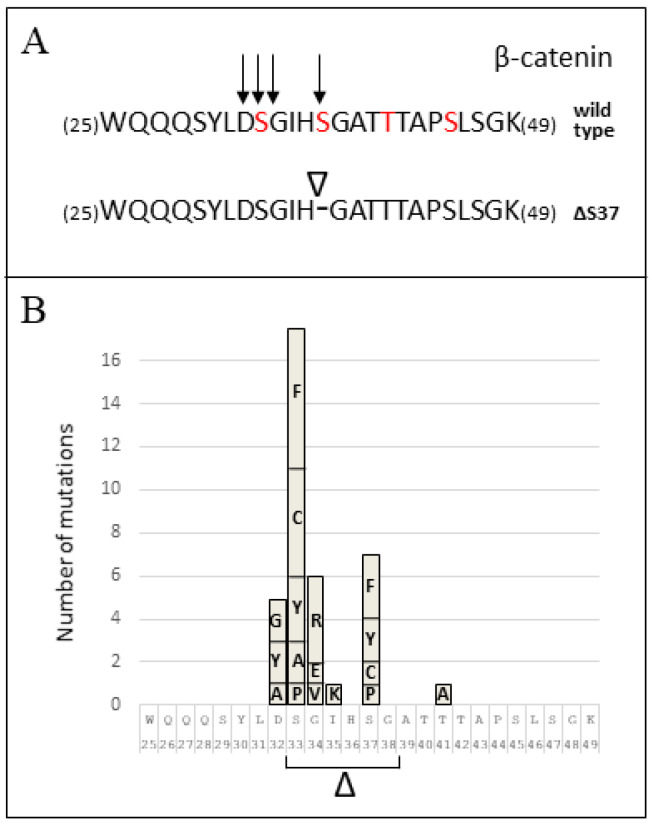
*CTNNB1* mutation distribution in medulloblastoma and functional characterization of the β-catenin ΔS37 variant. (**A**): The top line shows β-catenin amino acid composition between residues 25–49. Arrows indicate β-catenin residues mutated in our cohort of 88 pediatric medulloblastoma. Residues in red are regulatory phosphorylated residues. The bottom line shows the novel p.Ser37del β-catenin variant (ΔS37) found. Amino acids are denoted using the one-letter code. (**B**): The plot showing the identity and number of *CTNNB1* mutations found in pediatric medulloblastoma. Data are from Saint Jude pecan database.

**Figure 4 cancers-14-00421-f004:**
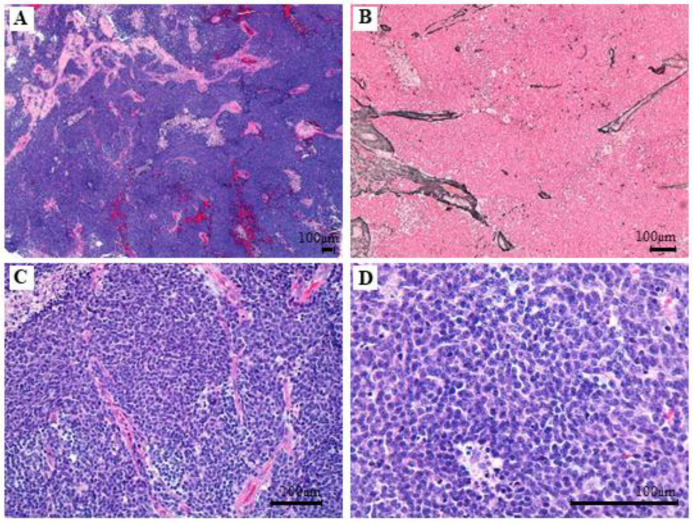
Hematoxylin-eosin staining of tumor specimen. Histological sections display tumoral (**A**): Hematoxylin-eosin stain, 40× magnification and no reticulin net enhancement (**B**): Reticulin stain, 100× magnification. No anaplastic or large cell changes are noticeable. (**C**): Hematoxylin-eosin, 200× magnification and (**D**): Hematoxylin-eosin, 400× magnification.

**Figure 5 cancers-14-00421-f005:**
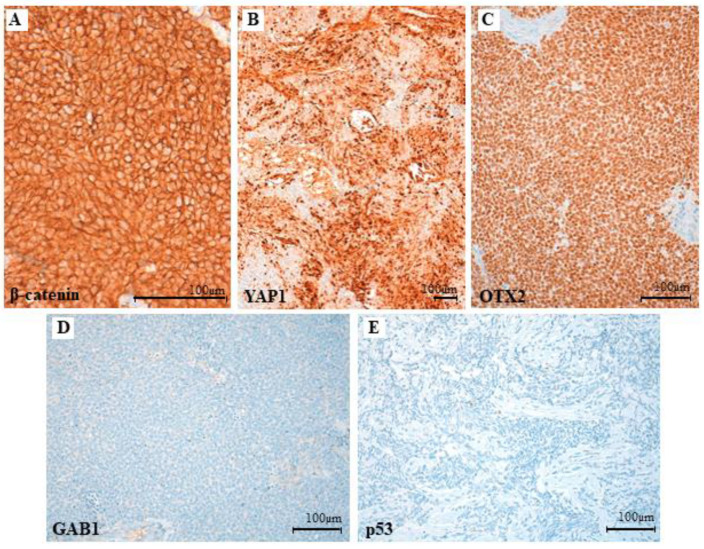
Immunostaining of tumor specimen. The immunoprofile shows patchy nuclear staining for β-catenin (**A**): 400× magnification, diffuse nuclear stain for YAP1 (**B**): 100× magnification and OTX2 (**C**): 200× magnification, and negative staining for GAB1 (**D**): 200× magnification and p53 (**E**): 200× magnification.

**Figure 6 cancers-14-00421-f006:**
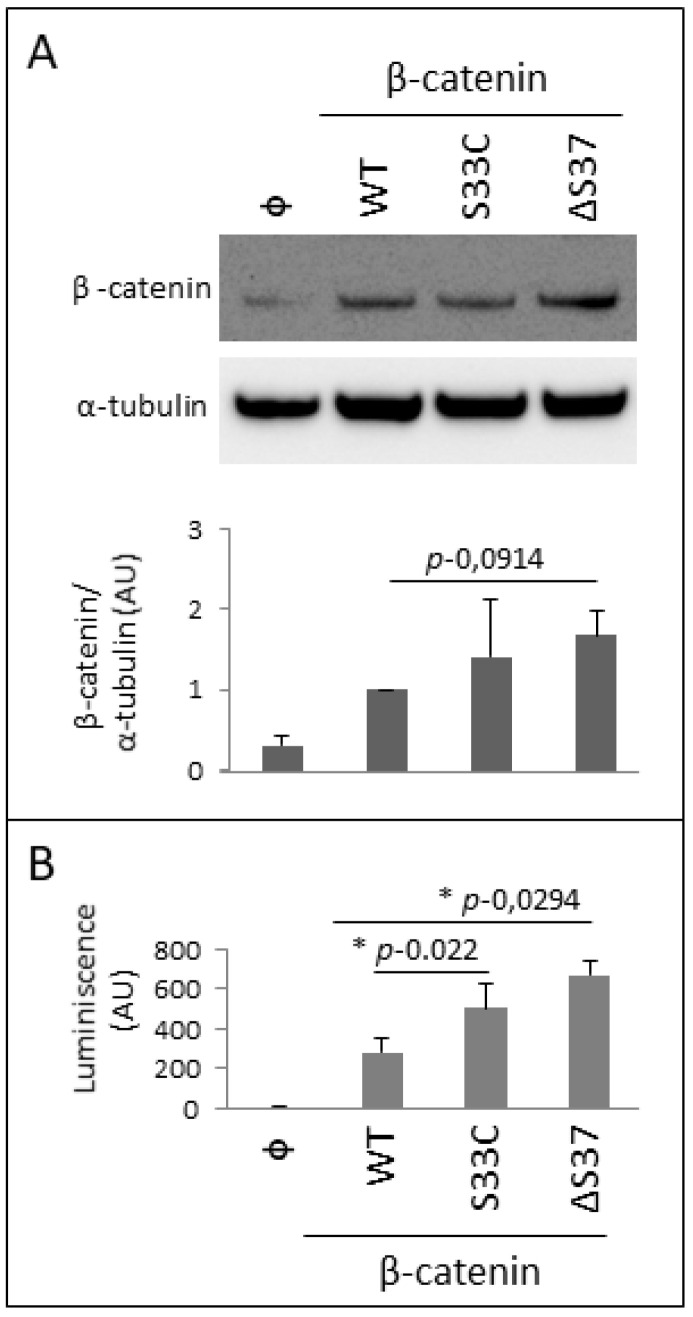
Functional characterization of the novel β-catenin variant c.109-111del (p.Ser37del). (**A**): Top panel. Immunoblot of endogenous β-catenin (φ), and recombinant β-catenin wild-type (WT) and S33C and ΔS37 variants. U2OS cells were transfected with empty vector (φ) or with plasmids containing the indicated β-catenin variants. Cell lysates were resolved on 4–10% SDS-PAGE under reducing conditions followed by immunoblot using anti-β-catenin antibody. Anti-α-tubulin was used to monitor protein loading. Bottom panel. Plot showing β-catenin/α-tubulin ratio, in arbitrary units (AU), from quantified immunoblot bands from two independent experiments ± SD. (**B**): Transcriptional activity of β-catenin variants. SEAP-normalized TCF/LEF-driven luciferase activity of β-catenin from U2OS transfected cells, as described in top panel. Luminescence is shown in arbitrary units (AU), from three independent experiments. Statistically significant results (*p* ˂ 0.05) are marked with *.

**Table 1 cancers-14-00421-t001:** *CTNNB1* mutations found in medulloblastoma tumor samples and in our study.

Amino Acid Position ^1^	Mutation ^2^	This Study	Number of MB Cases ^3^ PeCan COSMIC BioPortal			Mutation Type
32	c.94G>A/p.(Asp32Asn)/D32N c.94G>T/p.(Asp32Tyr)/D32Y c.94G>C/p.(Asp32His)/D32H c.95A>C/p.(Asp32Ala)/D32A c.95A>G/p.(Asp32Gly)/D32G c.95A>T/p.(Asp32Val)/D32V	- 1 - - 1 -	- 2 - 1 2 -	29 34 4 2 7 10	1 - - 1 2 -	Substitution-Missense Substitution-Missense Substitution-Missense Substitution-Missense Substitution-Missense Substitution-Missense
33	c.97_114del/p.(S33_G38del)/ΔS33-G38 c.97T>G/p.(Ser33Ala)/S33A c.97T>C/p.(Ser33Pro)/S33P c.98C>G/p.(Ser33Cys)/S33C c.98C>T/p.(Ser33Phe)/S33F c.98C>A/p.(Ser33Tyr)/S33Y	- - - 1 2 2	1 2 1 5 6 3	- 3 5 45 40 22	1 - - 4 6 2	Deletion-In frame Substitution-Missense Substitution-Missense Substitution-Missense Substitution-Missense Substitution-Missense
34	c.100G>A/p.(Gly34Arg)/G34R c.101G>A/p.(Gly34Glu)/G34E c.101G>T/p.(Gly34Val)/G34V	1 - -	4 2 1	25 17 6	5 1 1	Substitution-Missense Substitution-Missense Substitution-Missense
35	?/p.(Ile35Lys)/I35K c.104T>G/p.(Ile35Ser)/I35S	- -	1 -	1 1	- -	Substitution-Missense Substitution-Missense
37	c.109_111del/p.(Ser37del)/ΔS37 c.109T>C/p.(Ser37Pro)/S37P c.110C>G/p.(Ser37Cys)/S37C c.110C>T/p.(Ser37Phe)/S37F c.110C>A/p.(Ser37Tyr)/S37Y	1 - - - -	- 1 1 3 2	- 6 10 17 10	- 1 1 1 1	Deletion-In frame Substitution-Missense Substitution-Missense Substitution-Missense Substitution-Missense
40	c.119C>G/p.(Thr40Ser)/T40S	-	-	1	-	Substitution-Missense
41	c.121A>G/p.(Thr41Ala)/T41A	-	1	2	1	Substitution-Missense
45	c.134C>T/p.(Ser45Phe)/S45F	-	-	2	-	Substitution-Missense

^1^ Nucleotide and amino acid numbering are according to accessions NM_001904 and NP_001895. ^2^ Mutations are indicated following HGVS recommended nomenclature, as well as with single-letter code amino acid nomenclature. ^3^ Data from this study, St Jude PeCan (https://pecan.stjude.cloud, accessed on 1 December 2021), COSMIC (https://cancer.sanger.ac.uk, accessed on 1 December 2021), and cBioPortal (https://cbioportal.org, accessed on 1 December 2021).

## Data Availability

The data presented in this study can be available on request from the corresponding author. The data are not publicly available because they correspond to pediatric cancer patients as part of their clinical management.

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
