# Peer review of "Identification and Functional Analysis of a Novel *CTNNB1* Mutation in Pediatric Medulloblastoma"

_cancers, 2022, doi:10.3390/cancers14020421_

Round 1

Reviewer 1 Report

The authors confirm the occurrence of CTNNB1 mutations in WNT medulloblastomas (MB) and identify its additional novel form. This information is useful for further medical tests and treatment strategy selection. 

However, some of the cited materials stating mutation landscape status in MB are already out-of-date, e.g. latest is 2014. Possible solution to also mention/inspect latest studies e.g. https://doi.org/10.1038/nature22973  Similarly, the full medulloblastoma classes also have some updates, especially taking into account the latest WHO 2021 CNS tumor classification. This could be stated based on citation e.g. https://doi.org/10.1093/neuonc/noab106  

Further, the technical details of mutation calling and analysis are lacking in Methods section, only ABI sequencing procedure is stated. How was the identification/annotation/visualization of mutations from this data performed? 

Additional comments:

Table 1: lacks legend details. What tumor types are represented in identified mutations at columns "Number of cases"? Are there MBs inclusive? This could be stated clearly as additional value in the same column. 

Were any other classification methods for tumors except histology used for described cohort? This probably should be stated.

Reviewer 2 Report

In this manuscript, Alaña et al., investigated a cohort of 88 pediatric medulloblastoma tumors for exon 3 mutations from CTNNB1 gene, from which they identified genetic mutations from 9 cases, such as 8 missense point-mutations and 1 frame deletion mutation. The newly identified CTNNB1 in frame deletion (c.109-111del, pSer37del, ΔS37) is present in a pediatric patient with a classic medulloblastoma, WNT-activated grade IV (WHO 2016). By ectopically expressing ΔS37 β-catenin in U2OS human osteosarcoma cells, the authors found that ΔS37 β-catenin acquires enhanced stabilization and nuclear accumulation in comparison to its wild type counterpart. The paper is well written and the results are convincing. The reviewer would suggest its publication in Cancers with the following issues being addressed.

  1. Histology and Immunohistochemistry of molecular groups of pediatric medulloblastomas may be given to better showcase the difference.
  2. Have the authors confirmed that the anti-β-catenin antibody was able to detect all the wild 283 type β-catenin, ΔS37 and S33C β-catenin variants?
  3. There is no Fig 6A listed in the caption of Figure 6.
  4. Some typo and grammar errors. For example,

Line 138 The PCR reactions were carried out with of the following primer pair

Line 272 “and 1D: Hematoxylin-eosin, 400x magnification).”
